# Research Progress on Synthesis and Application of Cyclodextrin Polymers

**DOI:** 10.3390/molecules26041090

**Published:** 2021-02-19

**Authors:** Yuan Liu, Ting Lin, Cui Cheng, Qiaowen Wang, Shujin Lin, Chun Liu, Xiao Han

**Affiliations:** College of Biological Science and Engineering, Fuzhou University, Fuzhou 350108, China; N195720006@fzu.edu.cn (Y.L.); lintingtt33@163.com (T.L.); wendyeveryday@163.com (Q.W.); linshujin32@163.com (S.L.)

**Keywords:** cyclodextrin polymers, synthesis, separation science, materials science, biomedicine

## Abstract

Cyclodextrins (CDs) are a series of cyclic oligosaccharides formed by amylose under the action of CD glucosyltransferase that is produced by Bacillus. After being modified by polymerization, substitution and grafting, high molecular weight cyclodextrin polymers (pCDs) containing multiple CD units can be obtained. pCDs retain the internal hydrophobic-external hydrophilic cavity structure characteristic of CDs, while also possessing the stability of polymer. They are a class of functional polymer materials with strong development potential and have been applied in many fields. This review introduces the research progress of pCDs, including the synthesis of pCDs and their applications in analytical separation science, materials science, and biomedicine.

## 1. Introduction

Cyclodextrins (CDs) [1] is the general term for a series of cyclic oligosaccharides produced by amylose under the action of enzymes that are produced by Bacillus. They usually contain 6 to 12 glucopyranose units, and natural CDs are divided into α-CD, β-CD, and γ-CD with cavity sizes of ~0.5, 0.6, and 0.8 nm, respectively [2]. While single CD molecules can no longer meet the present practical application needs [3,4,5], the development of polymers has continued because of their excellent properties. They have become an important field of materials research and have brought new opportunities for CDs [6,7,8,9]. Studies involving CDs have demonstrated that they can also be used to form living polymers [10,11,12]. CD polymers (pCDs) can effectively address issues related to the manipulation of CD molecules, and they can endow unique functions and physical and chemical properties that are absent in single CD molecules. The advantages of the cavity structure of dextrin [13] include the simple formation of inclusions with guest molecules, control of the direction and rate of release of the guest molecules, and modifiability of the groups at the edge of the cavity; additionally, they can combine the excellent properties of polymers, including the mechanical strength and hardness, high relative molecular weight, and good thermal stability. In some cases, pCDs are also called cyclodextrin nanosponges (CD NSs). They do not appear all at once. They have undergone a long development. Since they were proposed in 1990, they have overcome the limitations of CDs, especially in water solubility. Great breakthroughs have been made in synthesis and application, and the form of cyclodextrins has been continuously changed in subsequent developments, especially in the past 50 years. The development of current pCDs started from a relatively simple cross-linking network in the 1960s, which was later developed into a multifunctional polymer [14]. Therefore, pCDs are widely used in various fields, including pharmaceutical, food, chemistry, chromatographic, catalysis, biotechnology, agriculture, cosmetics, hygiene, medicine, textile, and environmental fields [15,16,17,18].

This article introduces research progress in pCDs, including the synthesis of pCDs and their applications in analytical separation science, materials science, and biomedicine. It focuses on applications in biomedicine, and in particular, the technological innovations for application as drug delivery vehicles. Finally, the trends related to the development of pCDs are summarized and directions for future research are discussed.

## 2. Synthesis of pCDs

pCDs are polymer compounds containing CD units. They include crosslinked pCDs, linear pCDs, fixed pCDs, pCD inclusion compounds, and hyperbranched pCDs (Figure 1).

pCDs have different synthesis methods that depend on the final form. Crosslinked pCDs are obtained by pCDs and their derivatives using crosslinking agents with bifunctional or multifunctional groups. Commonly used crosslinking agents are citric acid, peroxides, isocyanates, acid anhydrides, and *N*,*N*-methylene bisacrylamide [19]. As an example, citric acid is a non-toxic crosslinking agent that has been used to graft α-CD onto cellulose fibers [20,21]. Ghorpade et al. [22] prepared a β-cyclodextrin-carboxymethyl cellulose (β-CD-CMC) hydrogel film via the esterification crosslinking method with citric acid for controlled release of ketoconazole (model drug). β-CD helps minimize the sudden release of the drugs. In crosslinked pCDs, the CDs are polymerized through the special functional groups of the crosslinking agent. The synthesis method is relatively simple, has strong operability, and can produce polymers with a high relative molecular weight, but the products have poor mechanical properties.

Linear pCDs are polymer compounds prepared from modified CD through alkenyl copolymerization or condensation polymerization of other special functional groups. In this process, the CD is first modified and then polymerized with almost no side reactions. Linear pCDs will not destroy the cavity structure of the CDs, so they have high potential for applications including for ion exchange, drug loading, separation, and adsorption. For example, the supramolecular linear polyacrylamide (SL-PAM) synthesized by You et al. [23] is a combination of β-CD and adamantane-terminated polyacrylamide (AT-PAM). It was prepared by the interaction between the host and the guest. SL-PAM samples were investigated using 2D NOESY NMR and thermal analysis to verify the formation of the inclusion compounds.

Fixed pCDs are polymer compounds formed by bonding CDs and its derivatives to a carrier. The material properties vary with the carrier, which includes inorganic polymers (such as silica gel and graphene) [24], natural polymers (such as cellulose and chitosan) [25], synthetic polymers (such as polystyrene, polyacrylate) [26]. For example, Shang et al. [27] synthesized an immobilized polyvinyl alcohol/CD ecological adsorbent and studied its application for removing ibuprofen from pharmaceutical wastewater. The adsorbent was prepared by solution blending (2-hydroxypropyl)-β-cyclodextrin (HPBCD) and polyvinyl alcohol (PVA), followed by glutaraldehyde treatment. The experimental process was simple and the product could be easily obtained. It can also be quickly reused via a simple soaking procedure. Immobilized pCDs have the advantages of good mechanical properties and a wide range of stable applications.

pCDs that form clathrates are produced by complexation between the polymer and the CDs. This inclusion compound has significantly better structure and properties than cyclodextrin and polymers. For example, CDs can improve the solubility of the guest [28,29,30]. Their synthesis methods include the saturated aqueous solution method, the ultrasonic method, grinding, colloid milling, freeze drying, and spray drying [31,32,33]. There are various synthetic methods that are widely used in the field of biomedicine and their applications will be explained in detail below. It is worth mentioning that the cyclodextrin polyrotaxane is a kind of pCD inclusion compound. Its structure includes a linear axis, multiple rings connected to the linear axis, and two end-capping groups connected to the linear axis. At each end, when the end of the linear shaft becomes larger than the inner diameter of the ring, or when molecules larger than the ring are bound to the end of the shaft, the ring on the polyrotaxane cannot be dissociated from the dumbbell-shaped shaft to make the polymer stably exist. The association constant K_a_ is a quantitative indicator reflecting the progress of the complex reaction. Angelina Angelova et al. [34] first reported a method for determining the association constant of amphiphilic water-soluble drugs. The amphiphilic peptide antibiotic polymyxin B (PMB) reacts with CD and assuming that CD and the compound drug are surface-inactive, and the two substances do not affect the surface properties of free PMB, the formula is calculated:(1)$$ka=\frac{C_{D}^{T}-\left[D\right]}{\left(C_{CD}^{T}-C_{D}^{T}+\left[D\right]\right)\left[D\right]}$$
where D represents PMB, $C_{D}^{T}$ and $C_{CD}^{T}$ respectively represent the total concentration of CD and PMB, [*D*] needs to use the concentration dependence of surface tension to be evaluated.

Further extended to water-insoluble drugs, the authors of [35] inferred the formula of the complexation of retinol (RL) and CD:(2)$$ka=\frac{{\left[RL\right]}_{T}-\left[RL\right]}{\left[RL\right]\left[CD\right]}$$

Among them, [*RL*] and [*CD*] represent the remaining interface concentration and the concentration of free CD molecules after RL molecules are exhausted. Here [*RL*]*_T_* represents the RL concentration before depletion. From this we can quantify the output of pCD, and this method is widely used in applications.

Similar to CDs, hyperbranched polymers also have a certain cavity structure and hydrophilic and hydrophobic properties. Some hyperbranched polymers have been used in the field of self-assembly [36]. The so-called hyperbranched polymer, i.e., a macromolecule with a highly branched structure, has the advantages of low viscosity and non-crystallinity. It has a highly branched structure with cavities and a large number of terminal functional groups. These characteristics give hyperbranched polymers the advantages of high solubility and reactivity [37]. Hyperbranched pCDs [38] have been developed on the basis of previous polymers, and are divided into three categories: (1) Bonding CDs to hyperbranched polymer; (2) complexing CDs to hyperbranched polymer, where the inclusion compound is formed on the polymer; (3) hyperbranched polymers synthesized with CDs as the core [38]. These methods combine the advantages of hyperbranched polymers and CDs, and have advantages such as good reactivity, high solubility, and broad application prospects.

Research needs to rethink traditional craftsmanship and request the latest synthesis methods. The new method is not mature enough, but with the development of the process, the yield and degree of polymerization will also become mature. It is worth noting that pCDs most commonly react with a suitable cross-linking agent in an organic polar aprotic solvent such as N, N-dimethylformamide (DMF) solution, but there will be some pollution. In recent years, people have been exploring solvent-free/green synthesis methods. Rubin Pedrazzo et al. proposed a green synthesis method through mechanochemical methods [39]. The test method is simple and the product is no different from the traditional organic solvent method. It is obtained by rotating anhydrous cyclodextrin and carbonyl diimidazole in a ball mill, washing with deionized water and acetone, and finally extracting. Max Petitjean et al. [40] cross-linked β-cyclodextrin-functionalized chitosan, xanthan gum, and locust bean gum to form a polymer under solvent-free conditions. The polymer has high stability, a large degree of crosslinking, and the method is simple, but homogenization of the solid mixture may occur. The article mentions that a small amount of water can be used to knead the mixture to prepare a paste solution, which has potential in the treatment of biologically active phenolic compounds, the purification of wastewater or the reuse of agricultural waste. Giancarlo Cravotto et al. [41] used low-boiling epoxy reagents in high-energy ball mills (HEBM) to simplify the preparation and purification of low-substitution (2-hydroxy) propylated β-and γ-cyclodextrins (β/γ-CDs). Compared with traditional methods, the properties of mechanically synthesized pCD, such as the degree of complexation, are different, and most of them are better. There are many such examples, which shows that the solvent-free/green synthesis method of pCDs, as a new direction, has attracted more and more attention and has great development potential.

The above text introduced the basic attributes and synthesis techniques for pCDs. So, what applications does the brand have in reality? The following mainly introduces research progress on pCDs from the latest applications in the fields of analysis and separation science, materials science, and biomedicine. It is worth noting that applications in the field of biomedicine, especially as a drug delivery system, have become a topic of intense research interest in recent years, and continued technological progress has also promoted the continuous development of pCDs and realization of their potential (Figure 2).

## 3. Application of pCDs in Analytical Separation Science

### 3.1. Application of pCDs in Wastewater Treatment and Water Purification

Over the past 30 years, water-related inorganic and organic micropollutants (such as heavy metals, drugs, and endocrine-damaging chemicals) are increasingly present in global water resources, and environmental issues have become a primary concern for society, public institutions, and industries [48,49,50,51,52]. The adsorption method has been extensively studied because it is economical, highly efficient, recyclable, and has good selectivity [53,54,55]. New pCDs can simultaneously adsorb and encapsulate a variety of organic and inorganic impurities, such as polycyclic aromatic hydrocarbons [56], pesticides [57], heavy metals [58], dyes [59], phenol compounds [60,61], phthalates [62], and pharmaceutically active compounds [63,64]. Hence, they have become topics of intense research interest because of their low cost and reusability. For example, β-CD/chitosan polymer was prepared by using glutaraldehyde as a cross-linking agent through solution polymerization, which could be developed as a new type of wastewater treatment purification material. Compared with traditional activated carbon, it has many advantages, such as renewability, low energy consumption, and low cost [65].

Laura et al. [66] studied the organic matter adsorption capacity of β-pCDs beads (BCPB) with different chemical compositions and thicknesses in solution. BCPB is a macromolecule produced by crosslinking β-CD with epichlorohydrin. They used a model solution containing ibuprofen and a total organic carbon (TOC) analyzer to determine the adsorption capacity. The results show that BCPB has excellent adsorption capacity for active organic drugs. Alsbaiee et al. [67] developed a porous CD derivative that crosslinked β-CD with rigid aromatics to form a high surface area mesoporous polymer. It was capable of quickly adsorbing a variety of organic impurities and it had 15 to 200 times the adsorption rate constant of traditional activated carbon. [68] Additionally, the polymer could be reused. Zhao et al. [42] reported a chitosan-EDTA-β-cyclodextrin (CS-ED-CD) multifunctional adsorbent prepared using EDTA as a crosslinking agent for the adsorption of toxic metals and organic trace pollutants in wastewater. The fixed CD cavity captured organic compounds while the EDTA-group complexed with the metal. This multifunctional adsorbent had improved potential for complex practical applications. The work provided new insights for the future design and preparation of sustainable materials for water purification. With continued development of CD and its derivatives, they are likely to become increasingly important in wastewater treatment and water purification, especially with the development of β-CD, which makes efficient water treatment possible (Figure 3). The above-mentioned relatively single treatment method has become increasingly unable to meet people’s needs. Scientists are exploring a non-polluting, efficient, and recyclable material to deal with pollution problems, and solar energy can be used to treat water pollution. Xuejiao Hu et al. [69] synthesized a new type of magnetic carboxymethylated β-CD-based porous polymer with fast adsorption performance and superparamagnetism in the water phase. The polymer has large pores and is adsorbed in the printing and dyeing wastewater through positive and negative electric attraction. The widespread anionic dyes are renewable materials with great potential. Garcia-Diaz et al. [70] developed a ROS-resistant fluorinated pCDs, which uses its adsorption capacity to adsorb pollutants near the catalyst, improves the utilization rate of photo-living oxygen, and optimizes the coating thickness on TiO_2_ microspheres. To improve the efficiency of pollutant degradation, the two microspheres combined to form CDP-TiO_2_ are expected to be used in photocatalytic water treatment. Sanaz Khammar et al. [71] grafted carboxymethyl-β-cyclodextrin (CM-β-CD) to the surface of core-shell titanium dioxide magnetic nanoparticles and successfully prepared recyclable CMCD-Fe_3_O_4_@TiO_2_, which is conducive to the adsorption of pollutants, protects nanoparticles, and promotes the photocatalytic activity of TiO_2_. Its cost-reduction, simple material, non-volatile and non-toxic properties have excellent application value in reducing the toxicity of polluted oil.

### 3.2. Application of pCDs in Analysis and Detection

Chirality is ubiquitous in nature [72,73] and it is very important in scientific research. For example, complex biological activity in the human body requires chirality [74,75]. However, the analysis and detection of chiral compounds is difficult [76]. Generally, achiral separation analysis of PMA and BMA [77,78,79,80,81,82] is not satisfactory. The unique cavity structure of pCDs and the molecule itself has multiple chiral centers, so it has good chiral resolution capabilities and can be used for analysis and detection. For example, by means of the different affinities between the β-CD unit and the two configurations of the racemate, enantioselective separation and detection of the racemate can be achieved.

Immohra et al. [83] applied the chiral characteristics of CD using the CD derivatives (hydroxypropyl-β-CD, CD oligomer, sulfobutyl ethyl ether)-β-CD, triacetyl-β-CD and hepta(2,6-di-*O*-methyl)-β-CD) identified as d- and l-glutamic acid, for the chiral recognition of t-asparagine, l-praziquantel, and its racemates. Ryvlin et al. [43] used the cavity structure of CDs and permethylated pCDs to detect and remove trichlorofluoromethane, which is harmful to the environment. The reaction produced a stable supramolecular molecule, which is a color transparent crystal complex that can be used repeatedly. Girschikofsky and Maiko [84] used permethyl, perethyl, and allallyl which substituted α-CDs, β-CDs, and γ-CDs as the sensitive sensor materials. Liang et al. [85] used the CD molecule itself to obtain multiple chiral centers, and they synthesized benzylureido-β via the reaction of 6-amino-β-cyclodextrin and reactive benzyl isocyanate. The -CD was bonded to silica gel through an addition reaction to obtain a new chiral stationary phase (BzCDP) based on benzylureido-β-cyclodextrin; it was successfully used to separate phenylthioglycolic acid (PMA) and benzyl mercapto acid (BMA) enantiomers, which have been shown to be biomarkers in human urine for benzene and toluene exposure. The separation of enantiomers has also been optimized through the study of their related factors. BzCDP is of great significance for the in-depth study of the presence and content of chiral markers in human urine, and for better understanding and evaluating of the harmful effects of benzenes on humans. Poor and Miklos [86] studied CDs and certain mycotoxins to form host–guest complexes, and removed alternan from aqueous solutions using insoluble β-CD bead polymers (BBP). Carcu-Dobrin et al. [87] studied the use of CD derivatives as chiral selectors to identify an optimized method for the chiral identification of amlodipine (AML) enantiomers. Carboxymethylethyl-β-CD (Cm-β-CD) was selected for enantiomeric identification. Through analysis and research, several factors were modified simultaneously to obtain an optimized separation method. Zheng et al. [88] used β-CD-gel and d- or l-tryptophan (homotype d-or l-Trp-gel) modified polyacrylamide-based gel for visible chiral recognition. In the NaCl aqueous solution, due to the obvious changes, the β-CD gel successfully distinguished the d- and l-Trp gels macroscopically, and the chirality difference becomes obvious, which will be very conducive to more in-depth research. It will also be more conducive to the understanding of chirality in the general public. The pCDs have shown their unique advantages in the field of analysis and testing in recent years, and they are a promising research direction.

## 4. Application of pCDs in Materials Science

### 4.1. Application of pCD Films

In recent years, membrane technology has been widely used in many fields of production because of its high separation efficiency, easy operation, and the absence of secondary pollution [89,90,91,92,93]. Compared with pCDs membranes, molecular sieves are expensive and have high energy consumption, and these issues are difficult to resolve. Because CD is produced at a large scale from starch, it has the advantages of being sustainable [94], non-toxic [95], and inexpensive; furthermore, it has been proven to be suitable for a variety of separations [96] and the production process is well established [97]. CD has been used as filler in films, a part of film-forming polymers, and as surface modifiers. pCDs membranes have excellent potential for applications, including in isomer separation and metal ion transport.

Villalobos et al. [44] studied the molecular level design of a new type of crosslinked CD filter membrane, which forms a CD film through interfacial polymerization. The filter membrane is cheap macrocyclic glucose with a shape similar to that of a hollow truncated cone (Figure 4). The channel-shaped cavity of the CD creates many pathways with a defined pore size in the separation layer, which can effectively distinguish molecules. The transport of molecules through these membranes is highly shape-sensitive. In addition, the cavities are hydrophobic and the ester-crosslinked outer part is hydrophilic, resulting in the high permeability of these membranes for polar and non-polar solvents (Figure 4).

Pangeniet et al. [98] used titanium glycinate-*N*,*N*-dimethylphosphonate to prepare cross-linked sulfonated polyvinyl alcohol membranes; then, they modified them by incorporating sulfonated β-CD. The ion exchange capacity of the membrane was found to be in the range of 1.40 to 2.55 meq/g. A high-precision impedance analyzer was used to evaluate the proton conductivity of the membrane at different temperatures and 100% relative humidity. The membranes containing 16% by mass and 20% by mass of sulfonated β-CD exhibited excellent proton conductivity of 0.121 and 0.143 S/cm at 80 °C, respectively. Wang, Yunze, and Lin et al. [99] proposed a new strategy to improve the flux and antifouling performance of ethylene vinyl alcohol (EVAL) membranes by blending with macrocyclic hyperamphiphiles (CD). A CD-rich layer was formed on the membrane surface. During the phase inversion process, the synergistic interaction between the hydrophobic and hydrophilic segments of the amphiphilic pCDs increases the membrane flux and increases the surface roughness and hydrophilicity of the membrane. In addition, the macrocyclic super amphiphilic hybrid membrane exhibited improved antifouling performance compared with the original EVAL membrane. The introduction of pCDs enabled the formation of a hydrophilic membrane surface, which has high application potential for practical membrane applications.

### 4.2. Application of CDs Functionalized Graphene Materials

The introduction of CD into the graphene family of materials is an important direction for graphene research. Graphene-based materials are widely used in macro/microstructures, sensors, oil/water separation membranes, and biomimetic interfaces [100,101,102,103,104]. CDs can improve their water solubility, biocompatibility, and supramolecular screening ability; hence, it may introduce new and interesting properties for these materials. CD-functionalized graphene materials have the properties of graphite, the inherent properties of olefins (high surface area, easy functionality [105], biocompatibility [106]), and the inherent properties of CDs.

Liu et al. [45] synthesized an excellent water-soluble nanosensor based on CD derivatives and graphene oxide; it was a supramolecular system in which the CD was loaded on the graphene oxide. The supramolecular system was very sensitive to Al^3+^, and the large specific surface area of the graphene oxide could capture Al^3+^. Simultaneously, the introduction of CDs could enhance the water solubility of graphene oxide, and this was the first self-assembled nanosensor composed of graphene oxide and CD derivatives. Its water solubility and excellent sensing activity effectively improved the application value of fluorescent nanosensors for tracking and detecting Al^3+^ in the environment and organisms. Chen et al. [107] developed a new type of β-CD with a large adsorption capacity and high throughput to effectively remove bisphenol A (BPA) (an environmental endocrine disruptor that can affect human health), i.e., a fine (β-CD) modified graphite oxide (CDGO) film. CDGO nanosheets are made by chemically grafting β-CD molecules to both sides of the GO nanosheets. The β-CD molecules can recognize and form stable complexes with BPA molecules to achieve efficient BPA removal, and the β-CD molecules on both sides of the CDGO nanosheets have a high grafting density, large surface area, and an interception efficiency of ~100%; its adsorption capacity is several times higher than the traditional method. Further, it could go through multiple operation cycles and is very promising for water treatment and molecular separation applications. Hu et al. [108] developed a host–guest recognition method using β-CD and Azo to prepare a new sandwich-type graphene/CD/C_60_ nanohybrid, which loaded β-CD through a one-pot reduction reaction. On graphene, it can be used to control the release of C_60_ and has better nitric oxide (NO) quenching ability than other graphene/C_60_ nanohybrids, which can serve as an effective nanoplatform against oxidative damage. The hybridization of rGO, β-CD, and Azo-C_60_ enhanced cell uptake and limited the aggregation of C_60_ and showed enhanced protection against NO-induced cytotoxicity. The rGO/β-CD platform can also be reused. Because host–guest chemistry and diazo chemistry are universal and generally applicable, this strategy can also be used to prepare other light-responsive nanohybrids, which should be valuable in the life science and materials science fields. Wang and Zhe [109] successfully synthesized β-CD functionalized three-dimensional graphene foam (CDGF) using a simple, one-step hydrothermal method. The effect of pH on the material was studied. Because the anion species of Cr(VI) are partially located on the positively charged surface of CDGF, when the pH of the Cr(VI) solution = 3, the CDGF has good selectivity for Cr(VI). As the pH increases, the adsorption capacity gradually decreases and the hydroxyl groups on CDGF play a major role in the adsorption process, which is a simple separation strategy. After adsorption of Cr(VI), CDGF maintains a fixed form and the separation process is simplified. This work provides a novel material for the adsorption of hexavalent chromium from water, and it provides direction for easy and fast solid–liquid separation strategies for adsorption and other applications (Figure 5).

## 5. Application of pCDs in Biomedicine

### 5.1. pCDs Reduce the Toxicity of Some Exogenous Organisms

The amorphous cavity of pCDs can capture a variety of drugs, thereby adjusting the physical and chemical properties of the guest molecule. Forming highly stable host–guest complexes with pCDs will reduce the biological effects of the guest molecules, and the improved bioavailability would reduce the toxicity of some exogenous organisms [110,111,112,113].

Guo et al. [46] prepared an inclusion compound of podophyllotoxin (PPT) and -CD, which could greatly reduce the toxicity of PPT. The behavior, characterization, and water solubility of the inclusion compound were carefully studied using multiple techniques. The inclusion compound was formed with a ratio of 1:1 and had a considerable stability constant K-s (4245.5 Lmol^−1^). The anti-cancer activity of the inclusion compound was better than that of cisplatin (DDP, positive control). Faisal et al. [114] studied the protective effect of -CD on the toxicity of HeLa cells and zebrafish embryos induced by zelaketone. Under certain conditions, sulfobutyl, methyl, and succinyl substituted CDs formed stable complexes, which significantly reduced or even eliminated the toxicity of azilenone. In addition, co-treatment with -CD also significantly reduced the sublethal effect of zaraketone. Studies have also shown that the formation of a stable zelenone-CD complex can significantly reduce or even eliminate the toxicity of zelenone in vivo and in vitro. Therefore, CD is expected to become a new mycotoxin binder.

### 5.2. Application of pCDs in Drug Delivery

The bioavailability of drugs is often limited by poor water solubility [115,116,117,118]. pCDs have been used for drug encapsulation, which improves the stability of drugs and effectively regulates their release. pCDs can greatly improve the solubility of poorly water-soluble drugs; additionally, they can also be used to prepare carrier systems that control drug release over a long time. CD has a special structure, which can enhance the biocompatibility and degradability of the drug by forming an inclusion compound, as well as a high loading rate of drug molecules and improvement of the controlled release performance. CDs can control the release of drugs and reduce their toxicity [119,120].

Oliveria et al. [47] used a high-yield reaction route based on polyglutamic acid, and used β-CD or γ-CD for the synthesis. The novel polymer had an average of ~17 CD cavities and was characterized using nuclear magnetic resonance, MALDI-MS, and DLS. It was determined to be a carrier of doxorubicin in human tumor cells, and this inclusion compound has antiproliferative activity in the tumor cells. Bisphosphonate is a mature drug with a wide range of applications in medicine. However, the side chain and nature of the phosphorus group may cause poor water solubility, thereby affecting its bioavailability. Mallard et al. [121] proved that CDs can be used as a bisphosphonate carrier. Through bisphosphonate functionalized CD, a cyclodextrin/bisphosphonate polymer (CD/BP) was synthesized and characterized. The formation of CD/BP was characterized by one-dimensional and two-dimensional nuclear magnetic resonance spectroscopy, isothermal titration, calorimetry, and ultraviolet-visible spectrophotometry, which showed that cyclodextrin is an effective carrier for bisphosphonates. CD/BP can be used to treat parasitic diseases, in particular, to prevent Chagas disease [122,123]. This provides a better treatment plan for the treatment of sleeping sickness caused by parasites. Ho et al. [124] proposed a submicrocarrier with an average hydrodynamic size of 400 to 900 nm through electrostatic gelation of anionic β-CD and chitosan (SMCs). This could address the issues of poor solubility of drugs and the limited bioavailability and pharmacological effects. It was used to improve the solubility and clinically relevant anti-infective controlled release properties of ciprofloxacin. In the study, it was found that when the encapsulation efficiency (~90%) and load capacity (~9%) of sMC were maximized, the molar ratio of ciprofloxacin to β-CD was 1:1. The results showed that regardless of their size, after 24 h of incubation, the cells absorbed sMC well without pathological changes. The sMC was non-toxic and had very good biocompatibility, and it is a suitable system with promising prospects for the treatment of extracellular lung infections. Khelghati et al. [125] focused on reducing the side effects of adriamycin by designing a pH-sensitive magnetic hyperbranched β-CD as a nano-level drug carrier. Nanoparticles were released more under acidic conditions and were released less under neutral alkaline conditions. The results of hemolysis analysis showed that the synthesized nanocarriers were completely biocompatible. In vitro studies have shown that free doxorubicin has a higher cytotoxicity than doxorubicin-loaded nanocarriers, demonstrating its high potential to deliver doxorubicin to tumor tissues. Li et al. [126] reported the design and synthesis of a new multifunctional nuclear initiator based on octa-norbornene functionalized γ-CD, which is an eight-arm star polymer [127,128] prepared by a nuclear-preferred ring-opening metathesis polymerization reaction. Hexaethylene glycol functionalized with norbornene was used to graft from the initiator. Using norbornene-functionalized polyethylene glycol (PEG) to extend the chain of the omega functional group to produce water-soluble diblock brush arm star copolymer (DBASC), the size of DBASC was between 10–11 nm, with very good thermal stability, long-range order, and crystallinity. Because of the introduction of γ-CD, DBASC has excellent solubility, enhanced drug binding ability, shows low toxicity, and has a strong inhibitory effect on MCF-7 breast cancer cells. Star polymers represent a new type of modular polymer platform with potential applications in nanostructure self-assembly and drug delivery. Das et al. [129] used β-CD as a drug delivery vehicle for drug cells with amino acid-based ionic liquid (AAIL) substitutes, which could improve the bioavailability of the therapeutic agent. Therefore, the use of β-CD is preferred to stabilize AAIL in our work, and AAIL can be further separated in a controlled manner. Because of its importance in the field of biotechnology, electrodialysis separation, and drug delivery, AAILs, i.e., proline nitrate PN and β-CD are proposed as a model system. In this work, the experimental measurements were theoretically related to quantum chemistry methods. Two-dimensional correlation experiments show the characteristics of PN and β-CD binding. These fascinating results were clarified with the help of molecular docking simulation studies. The confirmation with antibacterial research was consistent with the experimental results. The new AILs-based inclusion compounds have little toxicity and can be used as potential carriers. The results reported were encouraging for the practical application of AAIL and β-CD. Viale et al. [130] synthesized fibrin gel (FBG) and an amino-pCDs inclusion complex (oCD-NH_2_/Dox) in 2019, demonstrating that the FBG can be used in clinical or experimental applications to release different doxorubicin (Dox) nanoparticles. The fibrinogen (FG) and Ca^2+^ concentrations may change this activity. In vivo data support that the overall and local toxicity of FBG loaded with oCD-NH_2_/Dox is lower than that of FBG loaded with Dox. The results indicate that when administered locally via FBG loaded with oCD-NH_2_/Dox, the therapeutic index of Dox may increase, providing the possibility for using these delivery systems to treat neuroblastoma. Haley et al. [131] reported that a drug delivery system made of pCDs allowed for local administration of amphotericin B (AmB) (the leading drug for the treatment of clinical fungal infections), which can reduce toxicity to the host cells while maintaining the ability to eliminate fungal activity. By exploiting the molecular interaction between the CD cavity and the drug, a slow and sustained delivery rate of AmB was achieved. Lin [132] reported that β-cyclodextrin-{poly(lactide)-poly(2-(dimethylamino)ethylmethacrylate)-poly[oligo(2-ethyl-2-oxazoline)methacrylate]}_21_ [β-CD-(PLA-PDMAEMA-PEtOxMA)_21_] monomolecular micelles act as gold nanoparticles (GNP); the in situ formation and subsequent Dox encapsulated template were applied for the development of anti-cancer drug delivery and computer tomography (CT) imaging to form a pH-responsive therapeutically reactive nanocomposites in situ. Through a combination of experiments and dissipative particle dynamics (DPD) simulations, the formation, microstructure, and distribution of GNP and Dox were studied. Under acidic tumor conditions, Dox-loaded micelles had an encapsulation efficiency of 41–61%, showing rapid release (88% after 102 h). Both in vitro and in vivo experiments have shown excellent anti-cancer efficacy and effective CT imaging performance for β-CD-(PLA-PDMAEMA-PEtOxMA)_21_/Au/DOX. Single-molecule micelles represent a class of multifunctional nanocarriers for therapeutic diagnostics. Nanoparticle carriers are now a hot frontier field, and the intelligentization of drug loading can greatly increase chances of curing diseases such as cancer and NPC. In recent years, research on biodegradable nanocarriers has gradually increased, and pCDs have been used in new forms of drug delivery. Liu et al. [133] synthesized a new type of star-shaped nanocarrier (C_12_H_25_)_14_-β-CD-(SS-mPEG)_7_ (CCSP) for anti-tumor activity in 2019. The DOX-loaded nanocarrier CCSP has good biocompatibility, high drug loading, good stimulus response release performance, and low leakage, which have potential in anti-cancer intelligence. It is worth noting that Ran Namgung et al. [134] designed a new type of nano-assembled drug delivery system formed by the interaction between polymer-cyclodextrin conjugate and polymer-paclitaxel conjugate, which is the most popular CD polymer. This is one of the successful drug delivery systems. Nano-components have high stability, which can effectively deliver paclitaxel to targeted cancer cells through passive and active targeting mechanisms and effectively release them. Ying et al. [135] introduced the cationic CD loop into the multi-arm PEG backbone in a sterically selective manner, and developed a multi-arm pCD polyrotaxane nanocarrier platform that protectively encapsulates the interleukin 12 (IL-12) encoding plasmid for immune gene therapy of colon cancer. Compared with the linear pCD polyrotaxane, the multi-arm polymer design significantly improves the circulating half-life, and the reported tumor suppressor effect is excellent and non-toxic. Nowadays, pCD as an API is a hot topic in the field of biomedicine. For example, Atsushi Tamura et al. [136] have developed an acid-labile β-CD/Pluronic P123-based polyrotaxane for the treatment of a fatal metabolic disease, Niemann-Pick Type C (NPC), compared to the general 2-hydroxypropyl β-cyclodextrin (HP-β-CD) drug treatment, it can not only promote cholesterol excretion and prolong the life of the animal in animal models, and the required dose is greatly reduced, which has huge advantages compared with traditional medicine. In addition, in order to enhance the pharmacokinetics and biodistribution characteristics, and thereby improve the efficacy at lower doses, Aditya Kulkarn et al. [137] designed a β-CD-based polymer prodrug ORX-301, which is composed of β-CD two. It is formed by the polymerization of azide monomer and bifunctional ketal. It overcomes the main limitations of current β-CD-based NPC therapy, and is a potential alternative to existing treatment methods (Table 1).

## 6. Conclusions

pCDs have many advantages, including a sustainable source of raw materials, low cost, multiple synthetic methods, and a simple synthesis procedure. They not only have the characteristics of cyclodextrin molecules, such as an internal hydrophobic-external hydrophilic cavity structure and hand-shaped features, but they also have the excellent mechanical strength, hardness, and good thermal stability of polymers.

This article introduced the research progress of pCDs, including their synthesis and applications in analytical separation science, materials science, and biomedicine. The field of analysis and separation science mainly uses the cavity structure and hand shape characteristics of pCDs to realize separation and detection. In the field of materials science, CDs have been incorporated into various materials. Compared with traditional materials, pCDs have the characteristics of a high flux and solubility, resulting in practical materials with superior properties. In the field of biomedicine, compared with traditional materials, pCDs have the advantages of specific identification, non-toxicity, and good biocompatibility. In short, pCDs have been increasingly applied in the fields of analysis and separation science, material science, and biomedicine because of their unique characteristics.

While emphasizing the multi-field applications of pCDs, this article focused on its scientific and technological innovation as drug delivery vehicles, using the latest research as examples to illustrate their potential for cancer treatment and the elimination of fungi. A summary of the development trend in pCDs from multiple fields and perspectives illustrates the large growth in the amount of pCD research and the large potential for development, and this work provides direction for future research.

## Figures and Tables

**Figure 1 molecules-26-01090-f001:**
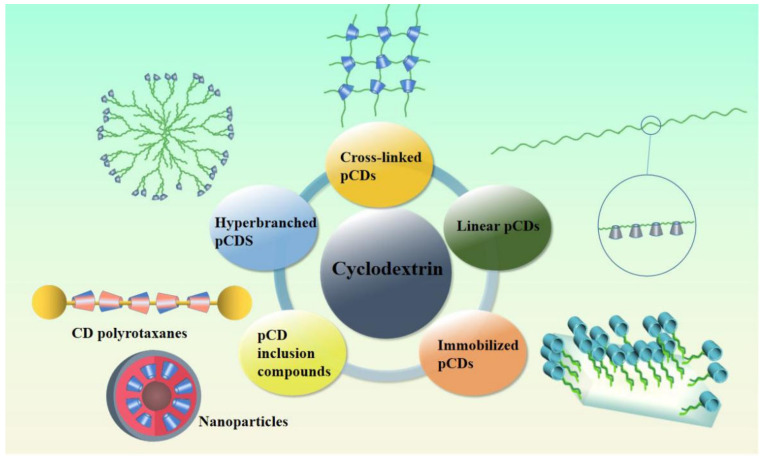
Five kinds of cyclodextrin polymers (pCDs) that containing cyclodextrin units.

**Figure 2 molecules-26-01090-f002:**
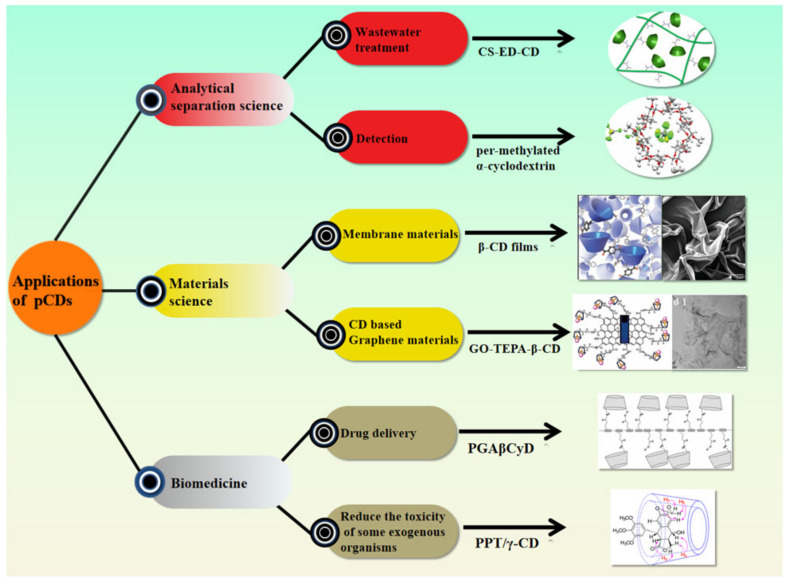
Multi-domain applications of CDs [42,43,44,45,46,47]. Adapted with permission from ref. [42]. Copyright 2017 Springer Nature; ref. [43]. Copyright 2018 John Wiley and Sons; ref. [44]. Copyright 2020 John Wiley and Sons; ref. [45]. Copyright 2018 Royal Society of Chemistry; ref. [46]. Copyright 2016 John Wiley and Sons; ref. [47]. Copyright 2017 Elsevier.

**Figure 3 molecules-26-01090-f003:**
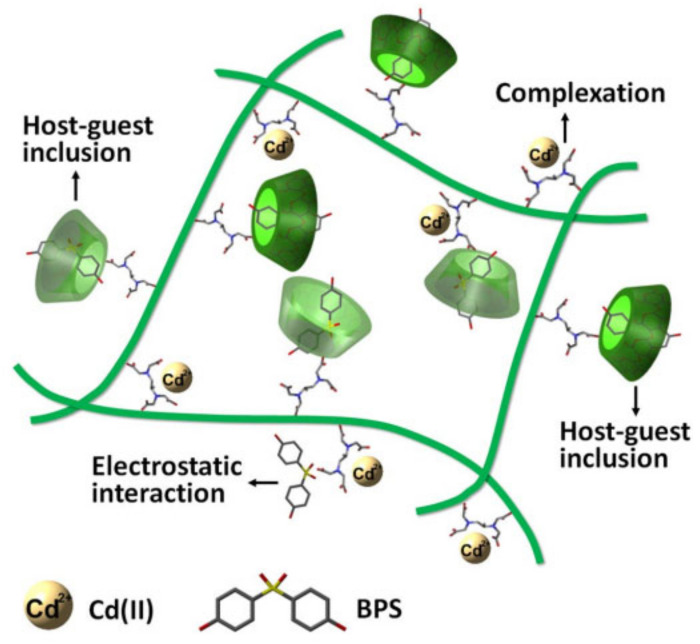
The schematic illustration of the related adsorption mechanisms of CS-ED-CD toward Cd(II) and BPS [42]. Adapted with permission from ref. [42]. Copyright 2017 Springer Nature.

**Figure 4 molecules-26-01090-f004:**
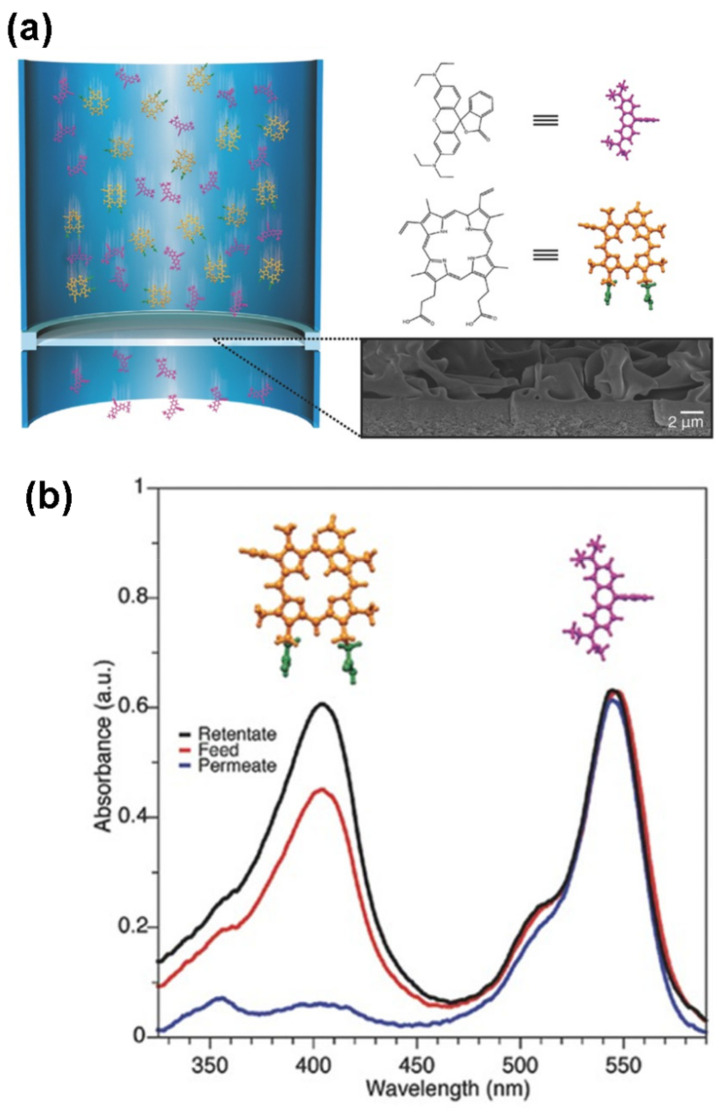
(**a**) Schematic showing how a β-CD membrane separates molecules based on their shape. Cross-section SEM image corresponds to a β-CD membrane prepared using 2 m NaOH aqueous solution. (**b**) UV–vis absorption spectra of a methanol solution with PPIX (orange molecule) and RB (pink molecule) to evidence the separation performance of the β-CD membrane [44]. Adapted with permission from ref. [44]. Copyright 2020 John Wiley and Sons.

**Figure 5 molecules-26-01090-f005:**
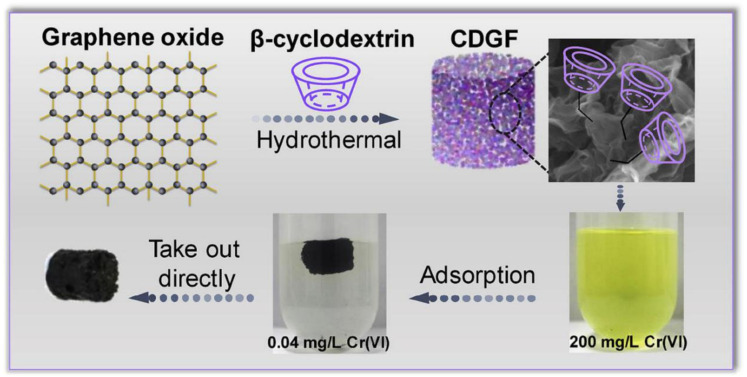
The β-CD functionalized three-dimensional structured graphene foam (CDGF) was applied for the adsorption of Cr(VI) with the easy and rapid separation strategy [109]. Adapted with permission from ref. [109]. Copyright 2019 Elsevier.

**Table 1 molecules-26-01090-t001:** Abstracts of papers for pCDs used as drug delivery vehicles.

Formulations	Drugs Carried	Application Effect	System Size	Ref.
PGAβCyD	Doxorubicin	Inhibit tumor tissue growth, Reduce the side effects of doxorubicin	5.5 nm	[47]
PGAγCyD.	Doxorubicin	Inhibit tumor tissue growth, Reduce the side effects of doxorubicin	5.5 nm	[47]
CD/BP	Bisphosphonate	Treat parasitic diseases	ND	[121]
sMC	Ciprofloxacin	Improve the solubility and release of the drug	400–900 nm	[124]
HPG-β-CD	Doxorubicin	Reduce the side effects of doxorubicin	30 nm	[125]
DBASC	Doxorubicin	Inhibit tumor tissue growth, Reduce the side effects of doxorubicin	10.0–11.0 nm	[126]
PN-β-CD	Real drug cell drugs	Non-toxic, drug-acceptable, low-cost, and environmentally friendly carrier	ND	[129]
CD-NH_2_/Dox	Doxorubicin	Treatment of neuroblastoma	3.2 nm	[130]
pCDs	AmB	Treatment of clinical fungus	2–15 kDa	[131]
β-CD-(PLA-PDMAEMA-PEtOxMA)_21_	Doxorubicin	Inhibit tumor tissue growth	27–28 nm	[132]
CCSP	Doxorubicin	Reduce drug leakage and increase drug load content	40–50 nm	[133]
Polymer—cyclodextrin conjugate	Polymer—paclitaxel conjugate	Confer high stability to the nano-assembly	54.6 ± 11.6 nm	[134]
pCD polyrotaxane	interleukin 12	Protective packaging	193 ± 6.2 nm	[135]
β-CD/Pluronic P123-based polyrotaxane	β-CD	Treating NPC	29,000 Da	[136]

## Data Availability

Data are available in a publicly accessible repository.

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
