# Peer review of "Research Progress on Synthesis and Application of Cyclodextrin Polymers"

_molecules, 2021, doi:10.3390/molecules26041090_

Round 1

Reviewer 1 Report

Submit the manuscript evaluation of [Molecules] Manuscript ID: molecules-1060315 This review focuses on the research progress of Cyclodextrin polymers, including synthesis methods, and potential applications in areas such as separation science, materials science, wastewater remediation, biomedicine, among others. However, I am struggling to see the novelty in this article, as a similar one (History of Cyclodextrin Nanosponges by Krabicová et al, Polymers MDPI) was published last year. References from 2018-2020 are also lacking in this review. My concerns regarding this manuscript are listed below: -The History of Nanosponges review (Krabicová et. Al., Polymers, MDPI, 2020) should be referenced in this review, as they address similar topics regarding synthesis and potential applications of pCDs. -Section 2 introduces the different synthesis methods of pCDs with different crosslinkers. As of 2019, solvent-free/green synthesis of pCDs have been reported. Those should be included in the review. -Section 3.1 introduces the potential application of pCDs in the removal of pesticides/pollutants from wastewater. The references on this section lack recent studies, as pCDs in many forms have been used for pesticide removal (including pCDs decorated with magnetic nanoparticles or pCDs with mechanical stability due to the presence of TiO2) -Section 5.2 introduces the application of pCDs in drug delivery. There are recent publications (2019-2020) addressing pCDs as biodegradable nanocarriers that should be cited in this review.

Author Response

This review focuses on the research progress of Cyclodextrin polymers, including synthesis methods, and potential applications in areas such as separation science, materials science, wastewater remediation, biomedicine, among others. However, I am struggling to see the novelty in this article, as a similar one (History of Cyclodextrin Nanosponges by Krabicová et al, Polymers MDPI) was published last year. References from 2018-2020 are also lacking in this review. My concerns regarding this manuscript are listed below:

Reply: Thanks a lot to the reviewer’s comments.

  1. -The History of Nanosponges review (Krabicová et. Al., Polymers, MDPI, 2020) should be referenced in this review, as they address similar topics regarding synthesis and potential applications of pCDs.

Reply: Thanks for the reviewer’s suggestion. In the revised manuscript,We have referenced The History of Nanosponges review. That is:

“In some cases, pCD is also called cyclodextrin nanosponges (CD NSs). It does not appear all at once. It has undergone a long development. Since it was proposed in 1990, it has overcome the limitations of CDs, especially in water-soluble Great breakthroughs have been made in sex and packaging, and the form of cyclodextrin has been continuously changed in subsequent developments. In the past, especially in the past 50 years, now pCD has started from a relatively simple cross-linking network in the 1960s. Development into a multifunctional polymer. [14] ”

  1. -Section 2 introduces the different synthesis methods of pCDs with different crosslinkers. As of 2019, solvent-free/green synthesis of pCDs have been reported. Those should be included in the review.

Reply: Thanks for the reviewer’s suggestion. In the revised manuscript,We have added the latest content about solvent-free/green synthesis of pCDs. That is:

“It is worth noting that pCDs are most commonly reacted with a suitable cross-linking agent in an organic polar aprotic solvent such as N, N-dimethylformamide (DMF) solution, but there will be some pollution. In recent years, people have been exploring solvent-free/green synthesis methods. Rubin Pedrazzo et al.[37] proposed a green synthesis method through mechanochemical methods in 2020. The test method is simple and the product is no different from the traditional organic solvent method. It is obtained by rotating anhydrous cyclodextrin and carbonyl diimidazole in a ball mill, washing with deionized water and acetone, and finally extracting. Max Petitjean et al.[38] cross-linked β-cyclodextrin-functionalized chitosan, xanthan gum and locust bean gum to form a polymer under solvent-free conditions. The polymer has high stability, a large degree of crosslinking, and the method is simple, but The homogenization of the solid mixture may occur. The article mentions that a small amount of water can be used to knead the mixture to prepare a paste solution, which has potential in the treatment of biologically active phenolic compounds, the purification of wastewater or the reuse of agricultural waste. Giancarlo Cravotto et al.[39]used low-boiling epoxy reagents in high-energy ball mills (HEBM) to simplify the preparation and purification of low-substitution (2-hydroxy) propylated β- and γ-cyclodextrins (β/γ-CDs). Compared with traditional methods, the properties of mechanically synthesized pCD, such as the degree of complexation, are different, and most of them are better.There are many such examples, which shows that the solvent-free/green synthesis method of pCDs, as a new direction, has attracted more and more attention nowadays and has great development potential.”

  1. -Section 3.1 introduces the potential application of pCDs in the removal of pesticides/pollutants from wastewater. The references on this section lack recent studies, as pCDs in many forms have been used for pesticide removal (including pCDs decorated with magnetic nanoparticles or pCDs with mechanical stability due to the presence of TiO2)

Reply: Thanks for the reviewer’s suggestion. In the revised manuscript,We have added the latest content about pCDs decorated with magnetic nanoparticles or pCDs with mechanical stability due to the presence of TiO2. That is:

“Xuejiao Hu et al. [67] synthesized a new type of magnetic carboxymethylated β-CD-based porous polymer with fast adsorption performance and superparamagnetism in the water phase. The polymer has large pores and is adsorbed in the printing and dyeing wastewater through positive and negative electric attraction. The widespread anionic dyes are renewable materials with great potential.Garcia-Diaz et al. [68] developed a ROS-resistant fluorinated pCDs, which uses its adsorption capacity to adsorb pollutants near the catalyst, improves the utilization rate of photo-living oxygen, and optimizes the coating thickness on TiO2 microspheres. To improve the efficiency of pollutant degradation, the two combined to form CDP-TiO2 microspheres are expected to be used in photocatalytic water treatment. Sanaz Khammar et al.[69] grafted carboxymethyl-β-cyclodextrin (CM-β-CD) to the surface of core-shell titanium dioxide magnetic nanoparticles and successfully prepared recyclable CMCD-Fe3O4@TiO2, which will It is conducive to the adsorption of pollutants, protects nanoparticles and promotes the photocatalytic activity of TiO2. Its cost reduction, simple material, non-volatile and non-toxic have excellent application value in reducing the toxicity of polluted oil.”

  1. -Section 5.2 introduces the application of pCDs in drug delivery. There are recent publications (2019-2020) addressing pCDs as biodegradable nanocarriers that should be cited in this review.

Reply: Thanks for the reviewer’s suggestion. In the revised manuscript,We have added the latest content about recent publications (2019-2020) addressing pCDs as biodegradable nanocarriers. That is:

“In recent years, research on biodegradable nanocarriers has gradually increased, and pCD has been used in new forms of drug delivery. Liu et al. [131] synthesized a new type of star-shaped nanocarrier (C12H25)14-β-CD-(SS-mPEG)7 (CCSP) for anti-tumor in 2019. The DOX-loaded nanocarrier CCSP has good biocompatibility, high drug loading, good stimulus response release performance and low leakage, which have potential in anti-cancer intelligence. It is worth noting that Ran Namgung et al. [132] designed a new type of nano-assembled drug delivery system formed by the interaction between polymer-cyclodextrin conjugate and polymer-paclitaxel conjugate, which is the most popular CD polymer. One of the successful drug delivery systems. Nano-components have high stability, which can effectively deliver paclitaxel to targeted cancer cells through passive and active targeting mechanisms and effectively release them.Ying et al.[133] introduced the cationic CD loop into the multi-arm PEG backbone in a sterically selective manner, and developed a multi-arm pCD polyrotaxane nanocarrier platform that protectively encapsulates the interleukin 12 (IL-12) encoding plasmid for colon cancer. Immune gene therapy. Compared with the linear pCD polyrotaxane, the multi-arm polymer design significantly improves the circulating half-life, and the reported tumor suppressor effect is excellent and non-toxic”

Reviewer 2 Report

This is the review of the manuscript entitled “Research progress on synthesis and application of cyclodextrin polymers”. In this review article, the authors introduced various cyclodextrin-based polymers and their applications in analytical separation sciences, material sciences, and biomedicine. The article is well written with referring new papers. Please refer following comments.

1) Totally, the article mainly includes introductions of research examples of CD polymers. The reviewer strongly recommends to add problems, tasks, comparison with other polymers, idea, suggestion of the authors, etc. to improve originality of this review article.

2) CD polyrotaxanes are representative CD polymers. The authors should add the syntheses and applications of CD polyrotaxanes.

3) The reviewer recommends to add following reference and brief mentions in 3.2 section. This is advanced and interesting research regarding chiral separation analysis using CD polymer. Yongtai Zheng, Yuichiro Kobayashi, Tomoko Sekine, Yoshinori Takashima, Akihito Hashidzume, Hiroyasu Yamaguchi, Akira Harada, Visible chiral discrimination via macroscopic selective assembly, Communications Chemistry volume 1, Article number: 4 (2018).

4) Applications of CD polymers as API (e.g. NPC) are hot topics in biomedicine field. Please refer and mention following studies. i) Aditya Kulkarni, Paola Caporali, Atul Dolas, Soniya Johny, Sandeep Goyal, Jessica Dragotto, Alberto Macone, Ramesh Jayaraman, Maria Teresa Fiorenza Linear Cyclodextrin Polymer Prodrugs as Novel Therapeutics for Niemann-Pick Type C1 Disorder, Scientific Reports volume 8, Article number: 9547 (2018) ii) AtsushiTamura, NobuhikoYui, Polyrotaxane-based systemic delivery of β-cyclodextrins for potentiating therapeutic efficacy in a mouse model of Niemann-Pick type C disease, Journal of Controlled Release, Volume 269, 10 January 2018, Pages 148-158

5) The following paper should be referred and mentioned in 5.2 section, because it is one of the most successful drug delivery systems using CD polymers. Ran Namgung, Yeong Mi Lee, Jihoon Kim, Yuna Jang, Byung-Heon Lee, In-San Kim, Pandian Sokkar, Young Min Rhee, Allan S. Hoffman, Won Jong Kim, Poly-cyclodextrin and poly-paclitaxel nano-assembly for anticancer therapy, Nature Communications volume 5, Article number: 3702 (2014).

6) The reviewer recommends to add some figures regarding their applications in biomedicine because there are no figures (only table).

7) RONDEL should be mentioned in biomedicine section, because it is one of the most successful examples of pCD.

Author Response

This is the review of the manuscript entitled “Research progress on synthesis and application of cyclodextrin polymers”. In this review article, the authors introduced various cyclodextrin-based polymers and their applications in analytical separation sciences, material sciences, and biomedicine. The article is well written with referring new papers. Please refer following comments.

Reply: Thanks a lot for the reviewer’s comments.

  1. Totally, the article mainly includes introductions of research examples of CD polymers. The reviewer strongly recommends to add problems, tasks, comparison with other polymers, idea, suggestion of the authors, etc. to improve originality of this review article.

Reply: Thanks a lot for the reviewer’s suggestion.We have added theproblems, tasks, comparison with other polymers, idea, suggestion . That is:

“Rethinking that the single inefficiency of removing water pollution by adsorption alone may also cause greater pollution, and it immediately leads to the characteristics that the materials encapsulated by pCDs can be recycled.”

“The above-mentioned relatively single treatment method has become increasingly unable to meet people's needs. Scientists are exploring a non-polluting, efficient and recyclable material to deal with pollution problems, and solar energy can be used to treat water pollution.”

“Reflect on traditional craftsmanship and demand the latest synthesis methods. In the article, the solvent-free/green synthesis is compared with the traditional method. The author feels that the new method is not mature enough, but with the progress of the process, the yield and degree of polymerization will also become mature.”

“It is worth noting that pCDs are most commonly reacted with a suitable cross-linking agent in an organic polar aprotic solvent such as N, N-dimethylformamide (DMF) solution, but there will be some pollution. In recent years, people have been exploring solvent-free/green synthesis methods. Rubin Pedrazzo et al.[37] proposed a green synthesis method through mechanochemical methods in 2020. The test method is simple and the product is no different from the traditional organic solvent method. It is obtained by rotating anhydrous cyclodextrin and carbonyl diimidazole in a ball mill, washing with deionized water and acetone, and finally extracting. Max Petitjean et al.[38] cross-linked β-cyclodextrin-functionalized chitosan, xanthan gum and locust bean gum to form a polymer under solvent-free conditions. The polymer has high stability, a large degree of crosslinking, and the method is simple, but The homogenization of the solid mixture may occur. The article mentions that a small amount of water can be used to knead the mixture to prepare a paste solution, which has potential in the treatment of biologically active phenolic compounds, the purification of wastewater or the reuse of agricultural waste. Giancarlo Cravotto et al.[39]used low-boiling epoxy reagents in high-energy ball mills (HEBM) to simplify the preparation and purification of low-substitution (2-hydroxy) propylated β- and γ-cyclodextrins (β/γ-CDs). Compared with traditional methods, the properties of mechanically synthesized pCD, such as the degree of complexation, are different, and most of them are better.There are many such examples, which shows that the solvent-free/green synthesis method of pCDs, as a new direction, has attracted more and more attention nowadays and has great development potential.”

“Nanoparticle carrier is now a hot frontier field, and the intelligentization of drug loading can greatly increase the cure of diseases such as cancer and NPC.”

  1. CD polyrotaxanes are representative CD polymers. The authors should add the syntheses and applications of CD polyrotaxanes.

Reply: Thanks for the reviewer’s suggestion. In the revised manuscript,We have supplemented the schematic diagram of CD polyrotaxanes in Figure 1, and applied many of the latest CD polyrotaxanes synthesis application examples:

“It is worth mentioning that the cyclodextrin polyrotaxane is a kind of pCD inclusion compounds. Their structure includes a linear axis, multiple rings connected to the linear axis, and two end-capping groups connected to the linear axis. At each end, when the end of the linear shaft becomes larger than the inner diameter of the ring, or when molecules larger than the ring are bound to the end of the shaft, the ring on the polyrotaxane cannot be dissociated from the dumbbell-shaped shaft to make the polymer stably exist.”

“Ying et al.[133] introduced the cationic CD loop into the multi-arm PEG backbone in a sterically selective manner, and developed a multi-arm pCD polyrotaxane nanocarrier platform that protectively encapsulates the interleukin 12 (IL-12) encoding plasmid for colon cancer. Immune gene therapy. Compared with the linear pCD polyrotaxane, the multi-arm polymer design significantly improves the circulating half-life, and the reported tumor suppressor effect is excellent and non-toxic.Nowadays, pCD as an API is a hot topic in the field of biomedicine. For example, Atsushi Tamura et al.[134] have developed an acid-labile β-CD/Pluronic P123-based polyrotaxane for the treatment of a fatal metabolic disease, Niemann-Pick Type C (NPC), compared to the general 2-hydroxypropyl β-cyclodextrin (HP-β-CD) drug treatment, it can not only promote cholesterol excretion and prolong the life of the animal in animal models, but also needs The dosage is greatly reduced, which has huge advantages compared with traditional medicine. In addition, in order to enhance the pharmacokinetics and biodistribution characteristics, and thereby improve the efficacy at lower doses”

  1. The reviewer recommends to add following reference and brief mentions in 3.2 section. This is advanced and interesting research regarding chiral separation analysis using CD polymer. Yongtai Zheng, Yuichiro Kobayashi, Tomoko Sekine, Yoshinori Takashima, Akihito Hashidzume, Hiroyasu Yamaguchi, Akira Harada, Visible chiral discrimination via macroscopic selective assembly, Communications Chemistry volume 1, Article number: 4 (2018).

Reply: Thanks for the reviewer’s suggestion. We have cited the references in the Application of pCDs in analysis and detection part. The cited references (ref. 86) are marked with blue color in the revised manuscript.

“Zheng et al.[86] used β-CD-gel and D- or L-tryptophan (homotype D- or L-Trp-gel) modified polyacrylamide-based gel for visible chiral recognition. In the NaCl aqueous solution, due to the obvious changes, the β-CD gel successfully distinguished the D- and L-Trp gels macroscopically, and the chirality difference becomes obvious, which will be very conducive to more in-depth research. It will also be more conducive to the understanding of chirality in the general public.”

86.Zheng, Y.; Kobayashi, Y.; Sekine, T.; Takashima, Y.; Hashidzume, A.; Yamaguchi, H.; Harada, A., Visible chiral discrimination via macroscopic selective assembly. Commun Chem 2018, 1.

  1. Applications of CD polymers as API (e.g. NPC) are hot topics in biomedicine field. Please refer and mention following studies. i) Aditya Kulkarni, Paola Caporali, Atul Dolas, Soniya Johny, Sandeep Goyal, Jessica Dragotto, Alberto Macone, Ramesh Jayaraman, Maria Teresa Fiorenza Linear Cyclodextrin Polymer Prodrugs as Novel Therapeutics for Niemann-Pick Type C1 Disorder, Scientific Reports volume 8, Article number: 9547 (2018) ii) AtsushiTamura, NobuhikoYui, Polyrotaxane-based systemic delivery of β-cyclodextrins for potentiating therapeutic efficacy in a mouse model of Niemann-Pick type C disease, Journal of Controlled Release, Volume 269, 10 January 2018, Pages 148-158

Reply: Thanks for the reviewer’s suggestion. We have cited the references in the Application of pCDs in drug delivery part. The cited references (ref. 134 and 135) are marked with blue color in the revised manuscript.

  1. Tamura, A.; Yui, N., Polyrotaxane-based systemic delivery of β-cyclodextrins for potentiating therapeutic efficacy in a mouse model of Niemann-Pick type C disease. J Control Release 2018, 269, 148-158.
  2. Kulkarni, A.; Caporali, P.; Dolas, A.; Johny, S.; Goyal, S.; Dragotto, J.; Macone, A.; Jayaraman, R.; Fiorenza, M. T., Linear Cyclodextrin Polymer Prodrugs as Novel Therapeutics for Niemann-Pick Type C1 Disorder. Sci Rep 2018, 8, (1), 9547.
  3. The following paper should be referred and mentioned in 5.2 section, because it is one of the most successful drug delivery systems using CD polymers. Ran Namgung, Yeong Mi Lee, Jihoon Kim, Yuna Jang, Byung-Heon Lee, In-San Kim, Pandian Sokkar, Young Min Rhee, Allan S. Hoffman, Won Jong Kim, Poly-cyclodextrin and poly-paclitaxel nano-assembly for anticancer therapy, Nature Communications volume 5, Article number: 3702 (2014).

Reply: Thanks for the reviewer’s suggestion. We have cited the references in the Application of pCDs in drug delivery part. The cited references (ref. 134 and 135) are marked with blue color in the revised manuscript.

 It is worth noting that Ran Namgung et al. designed a new type of nano-assembled drug delivery system formed by the interaction between polymer-cyclodextrin conjugate and polymer-paclitaxel conjugate, which is the most popular CD polymer. One of the successful drug delivery systems. Nano-components have high stability, which can effectively deliver paclitaxel to targeted cancer cells through passive and active targeting mechanisms and effectively release them.

  1. The reviewer recommends to add some figures regarding their applications in biomedicine because there are no figures (only table).

Reply: Thanks for the reviewer’s suggestion. We have modified the table and added data about the size of the drug delivery system to facilitate readers to understand the morphology of the material and the authenticity of the transport in the body. There will also be data in the literature cited in the article to facilitate readers’ intuitive understanding

  1. RONDEL should be mentioned in biomedicine section, because it is one of the most successful examples of pCD.

Reply: Thanks for the reviewer’s suggestion. In the revised manuscript,We have added the latest content about recent publications (2019-2020) addressing pCDs as RONDEL. That is:

“In recent years, research on biodegradable nanocarriers has gradually increased, and pCD has been used in new forms of drug delivery. Liu et al. [131] synthesized a new type of star-shaped nanocarrier (C12H25)14-β-CD-(SS-mPEG)7 (CCSP) for anti-tumor in 2019. The DOX-loaded nanocarrier CCSP has good biocompatibility, high drug loading, good stimulus response release performance and low leakage, which have potential in anti-cancer intelligence. It is worth noting that Ran Namgung et al. [132] designed a new type of nano-assembled drug delivery system formed by the interaction between polymer-cyclodextrin conjugate and polymer-paclitaxel conjugate, which is the most popular CD polymer. One of the successful drug delivery systems. Nano-components have high stability, which can effectively deliver paclitaxel to targeted cancer cells through passive and active targeting mechanisms and effectively release them.Ying et al.[133] introduced the cationic CD loop into the multi-arm PEG backbone in a sterically selective manner, and developed a multi-arm pCD polyrotaxane nanocarrier platform that protectively encapsulates the interleukin 12 (IL-12) encoding plasmid for colon cancer. Immune gene therapy. Compared with the linear pCD polyrotaxane, the multi-arm polymer design significantly improves the circulating half-life, and the reported tumor suppressor effect is excellent and non-toxic”

Reviewer 3 Report

Typically, review articles are great resources for introducing graduate students and established scientists to a new field. While the authors have done a good job highlighting potential field of applications for polymer cyclodextrins (pCDs), I do not think the authors have sufficiently dealth with the topic of pCD synthesis.

Instead of giving several examples of what has been done, this manuscript will be of greater benefit to readers if the authors can expatiate Section 2 to focus on pCD synthesis (methods/types, control, purification etc); or at the very least cite very good review articles that have dealt extensively with the topic of pCDs synthesis

Author Response

Typically, review articles are great resources for introducing graduate students and established scientists to a new field. While the authors have done a good job highlighting potential field of applications for polymer cyclodextrins (pCDs), I do not think the authors have sufficiently dealth with the topic of pCD synthesis.

Reply: Thanks a lot to the reviewer’s comments.

  1. Instead of giving several examples of what has been done, this manuscript will be of greater benefit to readers if the authors can expatiate Section 2 to focus on pCD synthesis (methods/types, control, purification etc); or at the very least cite very good review articles that have dealt extensively with the topic of pCDs synthesis

Reply: Thanks for the reviewer’s suggestion. In the revised manuscript,In the second part, we classify different polymers and explain the general synthesis method, and explain the polymer synthesis method of each cited article in the application. Among them, the green/solvent-free method of polymer synthesis introduces our attention. We cited a few examples and imagined the huge potential in the future. Researchers in the field of polymer chemistry tend to use cyclodextrin derivatives to develop the initial solutions. However, the synthesis is usually not optimal and lack of in-depth characterization of the product, but everyone has been developing selective and high-yield solutions. We have cited the references in the Application part. The cited references are marked with blue color in the revised manuscript.

  1. Rubin Pedrazzo, A.; Caldera, F.; Zanetti, M.; Appleton, S. L.; Dhakar, N. K.; Trotta, F., Mechanochemical green synthesis of hyper-crosslinked cyclodextrin polymers. Beilstein J Org Chem 2020, 16, 1554-1563.
  2. Petitjean, M.; Aussant, F.; Vergara, A.; Isasi, J. R., Solventless Crosslinking of Chitosan, Xanthan, and Locust Bean Gum Networks Functionalized with beta-Cyclodextrin. Gels-Basel 2020, 6, (4).
  3. Jicsinszky, L.; Calsolaro, F.; Martina, K.; Bucciol, F.; Manzoli, M.; Cravotto, G., Reaction of oxiranes with cyclodextrins under high-energy ball-milling conditions. Beilstein J Org Chem 2019, 15, 1448-1459.
  4. García-Díaz, E.; Zhang, D.; Li, Y.; Verduzco, R.; Alvarez, P. J. J., TiO2 microspheres with cross-linked cyclodextrin coating exhibit improved stability and sustained photocatalytic degradation of bisphenol A in secondary effluent. Water Res 2020, 183, 116095.
  5. Khammar, S.; Bahramifar, N.; Younesi, H., Preparation and surface engineering of CM-β-CD functionalized Fe3O4@TiO2 nanoparticles for photocatalytic degradation of polychlorinated biphenyls (PCBs) from transformer oil. J Hazard Mater 2020, 394, 122422.
  6. Liu, H.; Chen, J.; Li, X.; Deng, Z.; Gao, P.; Li, J.; Ren, T.; Huang, L.; Yang, Y.; Zhong, S., Amphipathic β-cyclodextrin nanocarriers serve as intelligent delivery platform for anticancer drug. Colloids and surfaces. B, Biointerfaces 2019, 180, 429-440.
  7. Namgung, R.; Lee, Y.; Kim, J.; Jang, Y.; Lee, B.-H.; Kim, I.-S.; Sokkar, P.; Rhee, Y.; Hoffman, A.; Kim, W., Poly-cyclodextrin and poly-paclitaxel nano-assembly for anticancer therapy. Nature communications 2014, 5, 3702.
  8. Ji, Y.; Liu, X.; Huang, M.; Jiang, J.; Liao, Y.-P.; Liu, Q.; Chang, C. H.; Liao, H.; Lu, J.; Wang, X.; Spencer, M. J.; Meng, H., Development of self-assembled multi-arm polyrotaxanes nanocarriers for systemic plasmid delivery in vivo. Biomaterials 2019, 192, 416-428.
  9. Tamura, A.; Yui, N., Polyrotaxane-based systemic delivery of β-cyclodextrins for potentiating therapeutic efficacy in a mouse model of Niemann-Pick type C disease. J Control Release 2018, 269, 148-158.
  10. Kulkarni, A.; Caporali, P.; Dolas, A.; Johny, S.; Goyal, S.; Dragotto, J.; Macone, A.; Jayaraman, R.; Fiorenza, M. T., Linear Cyclodextrin Polymer Prodrugs as Novel Therapeutics for Niemann-Pick Type C1 Disorder. Sci Rep 2018, 8, (1), 9547.